# Identification of Schizophrenia Susceptibility Loci in the Urban Taiwanese Population

**DOI:** 10.3390/medicina60081271

**Published:** 2024-08-06

**Authors:** Chih-Chung Huang, Yi-Guang Wang, Chun-Lun Hsu, Ta-Chuan Yeh, Wei-Chou Chang, Ajeet B. Singh, Chin-Bin Yeh, Yi-Jen Hung, Kuo-Sheng Hung, Hsin-An Chang

**Affiliations:** 1Department of Psychiatry, Tri-Service General Hospital, National Defense Medical Center, Taipei 114, Taiwan; 400010428@mail.ndmctsgh.edu.tw (C.-C.H.); light-0100@outlook.com (Y.-G.W.); fantine7520@ndmctsgh.edu.tw (T.-C.Y.); chinbinyeh@gmail.com (C.-B.Y.); 2Graduate Institute of Medical Sciences, National Defense Medical Center, Taipei 114, Taiwan; bett0125y@mail.ndmctsgh.edu.tw; 3Department of Radiology, Tri-Service General Hospital, National Defense Medical Center, Taipei 114, Taiwan; weichou.chang@gmail.com; 4IMPACT, The Institute for Mental and Physical Health and Clinical Translation, School of Medicine, Barwon Health, Deakin University, Geelong, VIC 3220, Australia; a.singh@deakin.edu.au; 5Division of Endocrinology and Metabolism, Department of Internal Medicine, Tri-Service General Hospital, National Defense Medical Center, Taipei 114, Taiwan; metahung@yahoo.com; 6Center for Precision Medicine and Genomics, Tri-Service General Hospital, School of Medicine, National Defense Medical Center, Taipei 114, Taiwan

**Keywords:** genetic loci, immunoglobulin-binding factors, schizophrenia, urban population

## Abstract

*Background and Objectives*: Genomic studies have identified several SNP loci associated with schizophrenia in East Asian populations. Environmental factors, particularly urbanization, play a significant role in schizophrenia development. This study aimed to identify schizophrenia susceptibility loci and characterize their biological functions and molecular pathways in Taiwanese urban Han individuals. *Materials and Methods*: Participants with schizophrenia were recruited from the Taiwan Precision Medicine Initiative at Tri-Service General Hospital. Genotype–phenotype association analysis was performed, with significant variants annotated and analyzed for functional relevance. *Results*: A total of 137 schizophrenia patients and 26,129 controls were enrolled. Ten significant variants (*p* < 1 × 10^−5^) and 15 expressed genes were identified, including rs1010840 (*SOWAHC* and *RGPD6*), rs11083963 (*TRPM4*), rs11619878 (LINC00355 and LINC01052), rs117010638 (*AGBL1* and MIR548AP), rs1170702 (LINC01680 and LINC01720), rs12028521 (*KAZN* and *PRDM2*), rs12859097 (*DMD*), rs1556812 (*ATP11A*), rs78144262 (LINC00977), and rs9997349 (*ENPEP*). These variants and associated genes are involved in immune response, blood pressure regulation, muscle function, and the cytoskeleton. *Conclusions*: Identified variants and associated genes suggest a potential genetic predisposition to schizophrenia in the Taiwanese urban Han population, highlighting the importance of potential comorbidities, considering population-specific genetic and environmental interactions.

## 1. Introduction

Schizophrenia is a profoundly intricate psychiatric disorder characterized by a range of clinical features, including a psychotic syndrome encompassing delusions, hallucinations, disorganized behavior, catatonic syndrome, and formal thought disorder, as well as negative symptoms like avolition, alogia, anhedonia, social withdrawal, and affective flattening [1,2]. Cognitive deficits in working memory, executive function, and processing speed, alongside comorbid substance use disorder and metabolic disease, are also recognized aspects of the condition [1,3]. Traditionally, the lifetime prevalence of schizophrenia has been approximated at 1%, with an equal impact on both sexes; however, some studies suggest a mean lifetime morbid risk of 1.19% [1,2,4]. The cumulative prevalence of schizophrenia among national health insurance enrollees in Taiwan approximates 0.64% [5]. Typically, schizophrenia emerges in late adolescence or early adulthood, prompting ongoing debates regarding its neurodegenerative or neurodevelopmental origins [6]. This condition contributes to significant societal burdens, challenges in achieving full recovery, long-term impairments in social and occupational function, and a heightened risk of reduced life expectancy due to unhealthy lifestyle habits, comorbidities, or an estimated 5% likelihood of suicide [1,2].

Schizophrenia, known for genetic heterogeneity, demonstrates heritability estimates of around 60–85%, as evidenced by well-established twin and family studies [2,6,7]. Over 200 distinct genomic loci and 100 genes are significantly associated with schizophrenia, revealing its polygenic nature characterized by common risk alleles with modest effects dispersed across the genome cumulatively [1,6,7]. Single-nucleotide polymorphisms (SNPs) are the most common type of genetic variation at the genome level, caused by a change in a single base pair within the DNA sequence. SNPs can be randomly found in either the coding or non-coding regions of genes, exerting corresponding effects on gene expression, mRNA processing, and protein translation.

Previous research has pinpointed noteworthy variants and implicated genes that regulate the immune response, notably within the major histocompatibility complex (MHC) region and specific genes involved in synaptic pruning, as well as markers in TCF4 and NRGN, implying implications in neurodevelopmental dysfunction and cognitive functioning in individuals with schizophrenia [7]. Given the role dopamine plays in schizophrenia, some studies implicate catecholamine O-methyltransferase (COMT) polymorphism, which is involved in dopamine metabolism. Specific SNP variants, especially rs4680 in the *COMT* gene, influence prefrontal cortical function with hypodopaminergic status and the manifestation of negative symptoms within acutely ill patients [8]. Additionally, research suggests that these variations might influence the response to antipsychotic medications in Asian schizophrenia patients [9]. Genes involved in encoding dopamine receptors and dopaminergic neurotransmission, such as *DRD2* and *AKT1/AKT3*, have also been associated with an increased schizophrenia risk [1,7,10]. Moreover, several genes are implicated in voltage-gated calcium channels and glutamatergic neurotransmission [1]. The collective evidence indicates that both common and rare genetic risk factors likely converge on the same underlying neuronal genes crucial for synaptic organization, differentiation, and transmission relevant to the pathogenesis of schizophrenia [1].

Functional annotation of significant variants allows comprehensive analysis across molecular pathways and gene sets, implicating gene expression, neurogenesis and neuron differentiation, synaptic transmission, and ion channel regulation [2,7]. Genes associated with schizophrenia risk are expressed specifically in hippocampal pyramidal cells, medium spiny neurons, and cortical interneurons [2]. Central nervous system-related gene sets include targets of fragile X mental retardation protein (FMRP), voltage-gated calcium ion channel complexes, and abnormal long-term potentiation, leading to disrupt synaptic protein function and neurodevelopment [7]. Further pathways also find lower levels of synaptic proteins, dendritic spines, gamma-aminobutyric acid (GABA)-ergic and glutamatergic markers, as well as disruption of complement-mediated synaptic elimination by microglia [2,7]. The involvement of pathways elucidates the underlying mechanisms of schizophrenia. Many implicated genes do not code for proteins but play a role in genetic processing, influencing transcription and various epigenetic factors.

While recent genome studies have identified numerous SNP loci associated with schizophrenia, offering a plethora of potential genetic variation resources with biological functions for the analysis of schizophrenia pathogenesis, most of them have been conducted in populations of European ancestry. Large-scale genetic studies in East Asian populations have generally shown highly similar results outside of the MHC region and SNP variant within the *GABBR1* gene, indicating dysfunction in the GABA system [7,11,12]. Even if genetic variants are similar across populations, the impact on risk might be specific to certain populations due to prominent but local contributions of clinical heterogeneity, gene–environment (GxE), or gene–gene (GxG) interactions [12]. Environmental exposures can induce gene regulatory changes associated with human diseases. In the context of schizophrenia, urbanization before birth or during early life is a risk factor for its development. Additionally, multi-factorial gene–environment interactions may play a significant role in the development of this disease. Supported by the social drift or selective migration hypothesis, individuals with a higher genetic load for schizophrenia risk often live in more densely populated urban environments, where they migrate and raise offspring, thereby passing on the higher genetic risk [13]. Identifying high-risk genes for schizophrenia in urban environments becomes particularly important.

Therefore, we aimed to identify susceptibility loci associated with schizophrenia, especially in individuals of Taiwanese Han ancestry in urban areas, utilizing data from the Taiwan Precision Medicine Initiative (TPMI). The main focus of this study is on SNP variants and associated genes with putative functional relevance in urban populations predisposed to schizophrenia.

## 2. Materials and Methods

### 2.1. Study Design, Participants and Ethical Approval

The Taiwan Precision Medicine Initiative (TPMI) is a collaborative project between Academia Sinica and 16 major medical centers in Taiwan, including the Tri-Service General Hospital (TSGH) [14]. Participants are recruited from various medical centers and genotyped by Academia Sinica. The project aims to create a comprehensive database containing detailed clinical and genetic profiles of one million individuals of Han Chinese ancestry in Taiwan. The study enrolled participants from TSGH, a major medical center in Taipei, the capital of Taiwan. Initially, individuals diagnosed with schizophrenia (ICD-10: F20.0, F20.1, F20.2, F20.5) were identified from TSGH TPMI participants based on Diagnostic and Statistical Manual of Mental Disorders, Fifth Edition (DSM-5) criteria through clinical psychiatric interviews. The control group consisted of individuals without schizophrenia spectrum disorder. The writing of this original article was approved by the Institutional Review Board of the Tri-Service General Hospital, following the informed consent of all the participating adults or legal guardians for incapacitated patients (IRB# 2-105-05-038).

Genomic DNA extraction involved collecting 3 mL of peripheral blood in EDTA vacutainers and subsequent purification using the QIAsymphony SP system (QIAGEN, Hilden, Germany). Each participant’s genomic type of purified DNA was loaded in the Taiwan Precision Medicine (TPM) array chip. Initially, extracted blood DNA was quantified using a NanoDrop spectrophotometer and then normalized. The normalized DNA was amplified and subsequently fragmented using restriction enzymes. The fragmented DNA was hybridized to probes on the TPM array chip, which was designed based on selected SNPs. After hybridization, unbound or non-specifically bound DNA was removed by washing the chip, and the hybridized DNA was stained with fluorescent dyes. The genome type signals from the stained TPM array were detected using an Axiom GeneTitan scanner (Thermo Fisher Scientific, Sunnyvale, CA, USA). For genotyping, the scanned fluorescent signal images were processed to extract raw intensity data. Subsequent quality control of the genome SNP data, genotype calling, and sample annotations were executed utilizing the Axiom Analysis Suite (Thermo Fisher Scientific, Sunnyvale, CA, USA) according to the user guide.

### 2.2. Schizophrenia-Genetic Association Analysis and Statistics

The steps conducted in the analysis are depicted in Figure 1. To analyze the relationship between schizophrenia and genomic variants, SNPs with a genotyping call rate below 80% on the TPM array were excluded. Subsequently, the remaining SNPs underwent association analysis utilizing the chi-squared method implemented in PLINK 1.9 software (https://www.cog-genomics.org/plink/: beta 7.2, accessed on 11 December 2023) [15].

Variants from the TPM array with low quality (minor allele frequency < 0.05 and Hardy–Weinberg equilibrium < 1 × 10^−6^) were removed. The filtered SNPs were further analyzed using a chi-squared test. The procedure for calculating the *p*-value obtained from the chi-squared test involved the following steps:Defined Phenotype Data: Participants diagnosed with schizophrenia were classified as the case group, while others were classified as control groups.Calculated Allele Counts: All allele counts, including reference and variant alleles from filtered SNPs (genotype data), were calculated for both case and control groups. These results were defined as observed counts.Derived Expected Counts: Based on the observed counts, expected counts were derived in a 2 × 2 contingency table, constructed from the number of reference alleles in cases, the number of variant alleles in cases, the number of reference alleles in controls, and the number of variant alleles in controls.Calculation of Chi-Squared Statistic: The chi-squared statistic was calculated using the expected and observed count values.Determination of Degrees of Freedom: The degrees of freedom for the chi-squared test were determined.Derivation of *p*-Values: *p*-values were derived from the chi-squared statistic and degrees of freedom.

Variants exhibiting high significance (*p*-values < 1 × 10^−5^) from the chi-squared test were selected for further characterization by annotation, such as genetic structure location, relative gene symbols, population frequencies, and biological functions. To estimate variant effect sizes, lasso regression (least absolute shrinkage and selection operator) [16] supported by PLINK 1.9 was applied:Preparation of Data: The phenotype and genotype data prepared were the same as those used in the chi-squared test.Loading Data into Lasso Model: Phenotype and genotype data were loaded into the lasso model, with the lambda parameter set to 0.95.Calculation of Effect Size: The effect size was calculated using lasso regression.

### 2.3. Variant Annotations and Functional Analysis

For variant annotations, encompassing gene symbols, the locations of coding or non-coding regions, and translated protein sequence alterations, wANNOVAR (https://wannovar.wglab.org/, accessed on 11 December 2023) [17] was utilized querying the RefSeq Database (https://www.ncbi.nlm.nih.gov/refseq/, accessed on 11 December 2023) supplied by the US National Institutes of Health (NIH). To compare variant allele frequency at a particular locus across diverse racial populations including African, American, Asian, and European, the allele frequency spectrum from the publicly available databases created in the 1000 Genomes Project (https://www.internationalgenome.org/, accessed on 11 December 2023) [18], Genome Aggregation Database (gnomAD) (https://gnomad.broadinstitute.org/, accessed on 11 December 2023) [19], and Taiwan BioBank (https://taiwanview.twbiobank.org.tw/, accessed on 11 December 2023) [20] were collectively merged for analysis.

Identified variant-associated genes underwent characterization of their biological functions and molecular pathways. Gene’s Biological Process (BP) in Gene Ontology [21], and related pathways in the Reactome pathway database [22] were interpreted by using Enrichr (https://maayanlab.cloud/Enrichr/, accessed on 11 December 2023) [23].

## 3. Results

### 3.1. Variants from Association Analysis

This study included 137 participants with schizophrenia and 25,927 control participants (Table 1). After conducting the schizophrenia-genetic association analysis, 280,153 SNPs were selected, all of which passed the filters for SNP call rate, minor allele frequency, and Hardy–Weinberg equilibrium. All variant analysis profiles are listed in Appendix A.

The analysis resulted in 10 significant variants (*p*-value < 10^−5^), including rs1010840, rs1170702, rs1556812, rs12028521, rs78144262, rs9997349, rs11083963, rs12859097, rs11619878, and rs117010638, with the Manhattan plot of the association analysis results displayed in Figure 2 and Table 2. All variants exhibited a high odds ratio (>1.5), except for rs12028521 positioned at chromosome 1, 14,175,667 bp, which notably presented a lower odds ratio (<1.0). The effect size of these variants indicates that rs78144262 had the strongest association influencing schizophrenia. Other variants, including rs11619878, rs12859097, rs11083963, rs1010840, and rs9997349, showed similar effects. In contrast, variants rs12028521 and rs117010638 had weaker effects, while rs1170702 and rs1556812 showed no effect.

The interpretation of association test quality in the Q–Q plot (Figure 3) revealed significant variant points positioned above the diagonal line, indicating a probable association with schizophrenia. Moreover, the lambda value from test statistics was close to 1, indicating an acceptable distribution of significance without significant false positives for analysis results.

The allele frequencies of most variants among controls within the 10 significant SNP variants closely resembled those observed in the East Asian and Taiwanese Han (TPMI and Taiwan Biobank) populations, distinguishing them from other human racial groups (as presented in Table 3). Notably, rs12028521, rs1010840, and rs117010638 have significantly higher allele frequencies in East Asian and Taiwanese populations compared to other racial groups (bolded in Table 3). Surprisingly, rs11619878 and rs1556812 show significantly lower allele frequencies in East Asian and Taiwanese populations than in other racial groups (underlined in Table 3), and when compared to the control group, these two rare SNPs exhibit higher allele frequencies in the case group of East Asian and Taiwanese Han populations. This suggests that these SNPs may be potential risk markers specific to Taiwanese urban Han populations associated with schizophrenia.

Based on the RefSeq annotation results, it was observed that the variants were associated with relative genes such as *AGBL1*, *ATP11A*, *DMD*, *ENPEP*, *KAZN*, *PRDM2*, *RGPD6*, *SOWAHC*, and *TRPM4* in the case group (Table 2). Although most of these variants are in the intergenic region between two neighboring protein-coding genes, they may contain regulatory DNA elements that lead to diverse functional gene expression. Importantly, the identified variants, specifically LINC00355, LINC00977, LINC01052, LINC01680, LINC01720, and MIR548AP, were situated in non-coding RNA, capable of regulating gene expression or translocation without encoding proteins. These variants and their associated transcripts could potentially act as high-risk factors for schizophrenia within the Taiwanese urban Han population. Further investigation in the GWAS Catalog database (https://www.ebi.ac.uk/gwas/, accessed on 11 December 2023) revealed that rs11619878 is associated with the degree of serum IgG glycosylation (Table 4), indicating potential involvement of immune system protein molecular glycosylation in schizophrenia, supporting the immune hypothesis of the disease.

### 3.2. Functional Annotations of Risk Genes

In Table 5, 75 significant (*p*-value < 0.05) Gene Ontology (GO) BP terms were identified, and the top 10 ones were presented in Figure 4. These genes’ biological function classes include bundle of His cell action potential (*TRPM4*), C-terminal protein deglutamylation (*AGBL1*), membrane depolarization during AV node cell action potential (*TRPM4*), peptide metabolic process (*ENPEP* and *DMD*), positive regulation of action potential (*TRPM4*), protein side chain deglutamylation (*AGBL1*), regulation of adipose tissue development (*TRPM4*), regulation of peptidyl-cysteine S-nitrosylation (*DMD*), regulation of skeletal muscle contraction (*DMD*), and regulation of systemic arterial blood pressure by renin-angiotensin (*ENPEP*). These biological functions are majorly involved in cardiac and skeletal muscle cell action, protein deglutamylation and nitrosylation, and regulation of arterial blood may have potential connections to schizophrenia.

### 3.3. Pathway Analysis

Pathway analysis revealed 10 significant pathways (*p*-value < 0.05) retrieved from the Reactome database (Table 6 and Figure 5): Carboxyterminal post-translational modifications of tubulin (*AGBL1*), ion channel transport (*ATP11A* and *TRPM4*), ion transport by P-type ATPases (*ATP11A*), metabolism of angiotensinogen to angiotensins (*ENPEP*), non-integrin membrane–ECM interactions (*DMD*), RHOF GTPase cycle (*SOWAHC*), sensory perception of sweet, bitter, and umami (glutamate) taste (*TRPM4*), sensory perception of taste (*TRPM4*), striated muscle contraction (*DMD*), and TRP channels (*TRPM4*). Most pathways are involved in cellular action. Of note, a few pathways are related to the biological function of neuronal dendritic spine formation (RHOF GTPase cycle), angiotensin peptide hormone metabolism impacting vasoconstriction and blood pressure elevation, and oral taste sensitivity. These results may support a potential association between genetic variants and symptoms accompanying schizophrenia. Further combined GO and pathway relationships revealed three distinct groups, including *ATP11A* and *TPM4* (Ion Channel), *DMD* and *ENPEP* (Blood pressure and Muscle Contraction), and *AGBL* (Tubulin glutamylation), sharing similar biological and molecular mechanisms (Figure 6).

## 4. Discussion

This study represents a pioneering effort in delineating schizophrenia risk loci within the Taiwanese urban Han population, uncovering nine genes and six non-coding RNAs closely associated with schizophrenia susceptibility. Among the identified genes (*AGBL1*, *ATP11A*, *DMD*, *ENPEP*, *KAZN*, *PRDM2*, *RGPD6*, *SOWAHC*, *TRPM4*) and non-coding RNAs (MIR548AP, LINC00355, LINC00977, LINC01052, LINC01680, LINC01720), the implications of the variants and their relevant genes underscore potential connections to schizophrenia etiology and related symptomatic presentations. These genes may be related to the onset, progression, prognosis, degeneration, and cognitive aspects of schizophrenia, as well as to the immune system and common comorbidities such as type 2 diabetes mellitus, hypertension, and alcohol use disorder. The results highlight the current state of this field and provide factors and potential explanations for future, more focused studies.

Notably, among these, the genetic variant rs11619878 has exhibited an association with serum IgG glycosylation levels, as indicated by the investigation in the GWAS Catalog database [24], suggesting a potential role of glycosylation in schizophrenia. The dynamic IgG glycosylation regulation is also associated with inflammation and autoimmunity-related diseases [25]. The conserved N-linked glycan in the second IgG domain of the heavy chain significantly impacts its structure and function [26]. Alterations in IgG N-glycans, such as galactosylation, fucosylation, and sialylation, intricately regulate diverse functions from inhibition to complement activation and cellular cytotoxicity. Galactose and sialic acid additions prompt anti-inflammatory responses, while core fucosylation modulates complement activity and antibody-dependent cellular cytotoxicity. Conditions like rheumatoid arthritis and aging exhibit shifts in IgG glycosylation, impacting inflammatory responses with increased fucosylation and reduced sialylation [26]. Reduced IgG galactosylation signifies various conditions, including inflammatory diseases (rheumatoid arthritis, systemic lupus erythematosus, Crohn’s disease, and nonalcoholic fatty liver disease), infections (*Mycobacterium tuberculosis*, HIV+, and HCV+), cancers (prostate cancer, gastric cancer), and aging, potentially influenced by IL-6 [26]. Interestingly, IL-6 affects brain structure and contributes to neuropsychiatric disorders like schizophrenia [27]. Schizophrenia patients display distinctive IgG glycosylation alterations, characterized by increased circulating IgG levels, the presence of a pro-inflammatory IgG receptor in the midbrain, and reduced glycosylation of glutamate transporters EAAT1 and EAAT2 in the prefrontal cortex, along with altered gene expression related to IgG N-glycan biosynthesis [28]. Moreover, this variant lies amidst two long non-coding RNAs: LINC00355 and LINC01052. Investigation of lncRNA expression in schizophrenia patients’ peripheral blood leukocytes might unveil trans-acting effects on mRNA expression [29].

Multiple studies have corroborated associations between specific genetic factors and schizophrenia. An analysis in the Russian population identified the AGBL1 gene as being linked to schizophrenia and cognitive abnormalities, characterized by tubulin glutamylation effects, which signifies its involvement in neuropsychiatric disorders [30]. Burbaeva et al. observed reduced colchicine-binding activity of tubulin in the prefrontal, limbic, and temporal cerebral cortex in schizophrenia patients [31]. Microtubule stabilizers such as NAP (davunetide) have been reported to normalize transcriptional deficiencies commonly observed in mouse models of schizophrenia, positioning them as potential candidates for clinical use to treat schizophrenia [32]. Additionally, research by Goetzl et al. revealed lower levels of plasma neuron-derived extracellular vesicles (NDEV), particularly for transient receptor potential cation channel subfamily M, member 4 (TRPM4), in first episodes of psychosis [33]. The reduction in NDEV levels of TRPM4, coupled with decreased stabilizing complexes involving NMDAR1, suggests a potential reduction in mitochondrial Ca^2+^ uptake, thereby raising the risk of neuronal toxicity [33]. In addition, human transient receptor potential (TRP) channels play a crucial role in various pathological conditions, including excitotoxicity [34]. Glutamate excitotoxicity is a central mechanism leading to cellular dysfunction and death in neuropsychiatric disorders such as schizophrenia [34]. Consequently, the significance of TRPM4 highlighted potential dysfunctions related to mitochondrial calcium homeostasis and glutamate excitotoxicity pathways, which may be implicated in the onset and progression of schizophrenia.

Schizophrenia patients also face an increased risk of developing type 2 diabetes mellitus (T2DM), a major factor in cardiovascular disease, even among medication-naive patients. The prevalence of T2DM in individuals with schizophrenia is 2 to 3 times higher than in the general population, ranging from 6% to 21%. The gene *ATP11A* with ion channel transport function exhibits a significant correlation with fasting glucose and HbA1c levels. Hypermethylation of *ATP11A* links to reduced expression, contributing to elevated fasting glucose and HbA1c levels, crucial in T2DM development [35]. Additionally, changes in *ATP11A* have implications beyond diabetes, potentially impacting disease-free survival rates in colorectal cancer, notably affecting schizophrenia patients, who face a 50% higher risk of cancer-related mortality [35,36]. The *ATP11A* genes might also underlie the genetic connection between schizophrenia and T2DM.

Patients with schizophrenia face a relatively high mortality rate among individuals with mental disorders, largely attributed to hypertension and limited engagement with preventive interventions, including blood pressure management [37]. The prevalence of hypertension in schizophrenia stands at 39%, driven by factors, including obesity, antipsychotic medication, inflammation, and irregular autonomic nervous system activity [38]. Holmes outlined the role of the ENPEP gene and its protein products, predominantly expressed in kidney glomerular and proximal tubule cells, in the renin-angiotensin system’s catabolic pathway, contributing to the production of angiotensin III. This compound significantly impacts blood pressure regulation, angiogenesis, and potentially contributes to hypertension risk [39]. Furthermore, studies indicated that underlying angiogenesis in the brain, via the vascular endothelial growth factor (VEGF) pathway, may influence brain blood flow regulation, implicating its role in schizophrenia’s mechanisms [40].

Schizophrenia and attention deficit co-occur in individuals with neuromuscular diseases like Becker muscular dystrophy, featuring “in-frame” *DMD* gene deletions [41]. The *DMD* gene on chromosome Xp21 encodes vital cytoskeletal protein dystrophin. Mutations in *DMD* underlie muscular dystrophies, affecting motor function, cognition such as spatial and recognition memory, and altering behavior, as seen in mice’s fear response [41]. The severity of neurodevelopmental changes varies based on the type and location of *DMD* gene mutations, and affected dystrophin isoforms [41]. DMD genes also intersect with depression and influence Alzheimer’s disease risk, suggesting a potential link to depressive symptoms, suicide rates, and cognitive deterioration in schizophrenia [42].

Additional newly identified variants in *KAZN*, *PRDM2*, and *RGPD6* genes within the Taiwanese Han population present intriguing implications. Periplakin interacting protein, called KAZN, is involved in chromosome assembly, cell adhesion, cytoskeletal organization, and epidermal differentiation [43]. *KAZN* has been implicated in patients with schizophrenia. For example, Belangero et al. reported methylation of the KAZN gene in schizophrenia prognosis, hinting at environmental risk factors triggering gene methylation and schizophrenia symptoms [44]. *PRDM2* gene products possess dual functions (i.e., binding to DNA and transcription factors) and potentially regulate numerous cellular genes [45]. Transcriptome analyses reveal significant PRDM2 expression enrichment in brain regions, particularly the prefrontal cortex (PFC) [45]. Intriguingly, alcohol-dependent rats exhibit downregulated PRDM2 expression following a history of alcohol dependence [45,46]. PRDM2 knockdown increases alcohol self-administration, aversion-resistant alcohol intake, and stress-induced relapse to alcohol seeking [46]. This aligns with schizophrenia’s high co-occurrence with alcohol use disorder, estimated at a lifetime prevalence of 24.3%. The *RGPD6* gene’s association with autism spectrum disorder and intellectual disability underscores potential common symptomatic manifestations across neurodevelopmental disorders [47].

This study has several limitations. First, our study failed to demonstrate some variants that were found in previous large-scale genetic studies in schizophrenia. The results reported herein should be interpreted with caution because the generalizability of the findings is restricted by the small sample size and unmeasured confounding factors in the population demographics. Second, the present study investigated common genetic variations with a minor allele frequency < 0.05 excluded. Therefore, there may be rare variants of importance related to schizophrenia that fall outside the scope of this study. Third, we must consider potential heterogeneity in the schizophrenia phenotype, particularly concerning positive and negative symptoms. Fourth, while risk variants are linked to schizophrenia, the concordance rate in monozygotic twins is only about 50%. This underscores the importance of non-genetic factors (such as methylation and microbiomes) and other environmental influences (such as obstetric complications) and their interactions with genetic factors in increasing schizophrenia risk. Additionally, while our study identified many variants and relative genes potentially comorbid with schizophrenia, the cross-sectional design limits the ability to determine whether the shared genetic influences found in this and previous studies are causal factors for the comorbidity of schizophrenia. Social and environmental factors are likely to play at least as large a role as genetic ones in the development and co-occurrence of schizophrenia. Fifth, since the study subjects were all from urban populations, future research should include both urban and rural populations to confirm the genetic uniqueness and interactions associated with urban living. Finally, the TPMI project began at TSGH in 2020, and patients who did not visit the outpatient department after 2020 were not included in this study.

## 5. Conclusions

In conclusion, the genetic predisposition to schizophrenia demonstrates a remarkable complexity within the Taiwanese urban Han population. The identification of novel variants underscores the substantial genetic involvement in schizophrenia’s pathogenesis, revealing intricate links between genetic variations, immune responses, metabolic irregularities, and the spectrum of neuropsychiatric manifestations. Further research is needed to clarify the relationship and the causation of risk variants in Han urban population schizophrenia.

## Figures and Tables

**Figure 1 medicina-60-01271-f001:**
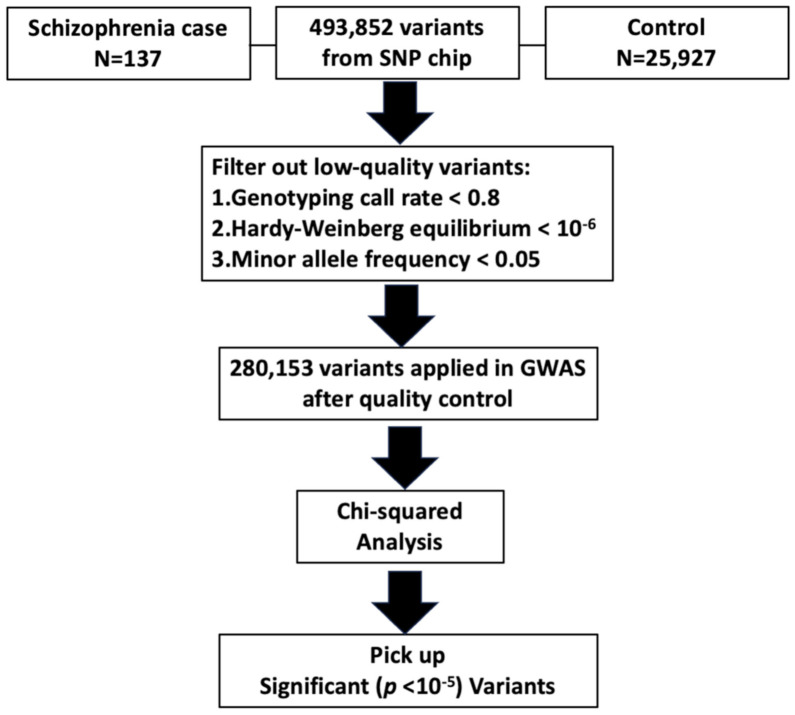
The study encompassed 137 schizophrenia patients as case groups and 25,927 participants as the control group. Genotyping was conducted using the Taiwan Precision Medicine (TPM) array chip. A dataset of 493,852 SNPs underwent filtration, with 280,153 SNPs passing through and subsequently subjected to chi−squared testing for the detection of risk factors.

**Figure 2 medicina-60-01271-f002:**
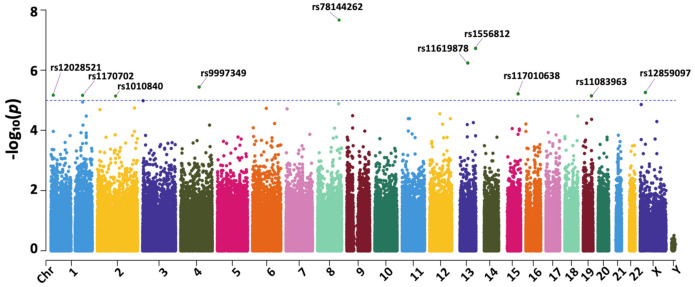
Manhattan plot of associated variants results in schizophrenia patients. Following chi−squared testing, 280,153 variants were detected among TSGH TPMI participants. Among these, ten highly significant variants were identified based on a *p*-value < 10^−5^ (green dots above the blue dashed line): rs78144262, rs9997349, rs1010840, rs11083963, rs11619878, rs1170702, rs117010638, rs12028521, rs12859097, and rs1556812.

**Figure 3 medicina-60-01271-f003:**
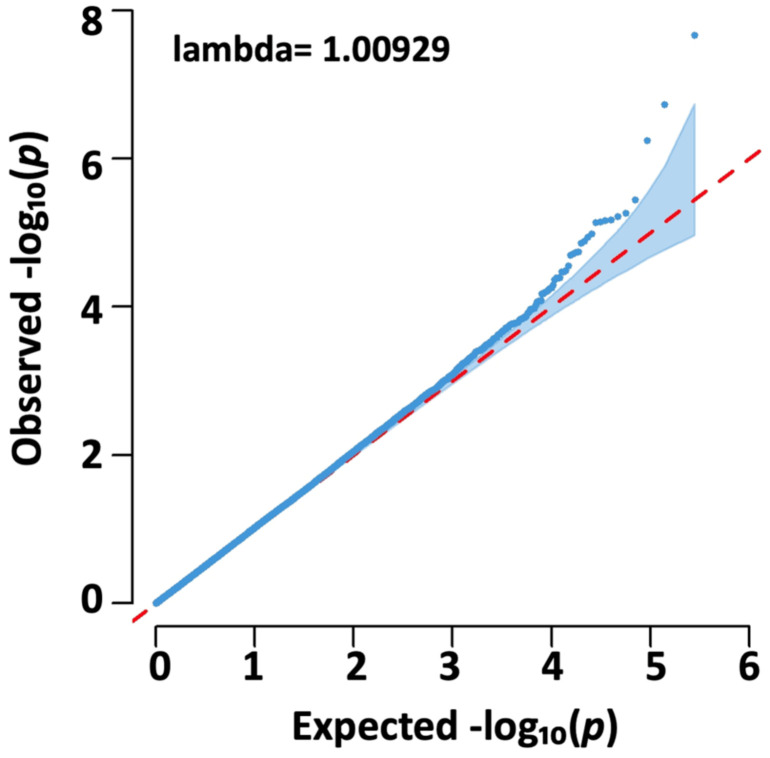
Interpretation associations results in Q-Q (quantile-quantile) plot. The distribution of low *p*-values (represented by high observed −log10(*p*) values) indicates true associations with schizophrenia. The genomic inflation factor, lambda, was determined based on the median observed and expected chi-square values.

**Figure 4 medicina-60-01271-f004:**
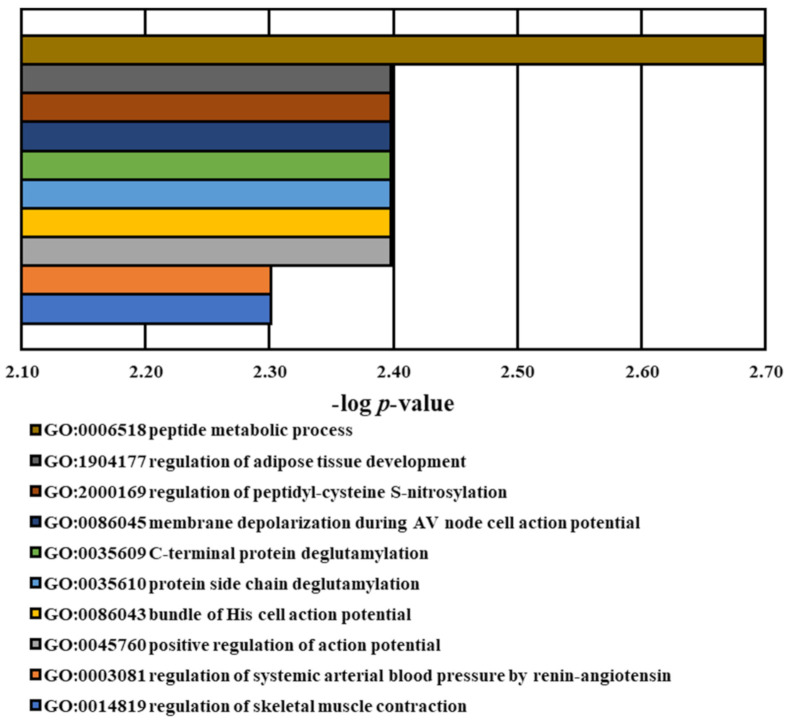
Top 10 significant Gene Ontology biological process (GO BP) functions relative to variants. The BP functions of genes associated with identified risk variants were characterized using the GO database. Selected GO terms exhibited with *p*-values < 0.05.

**Figure 5 medicina-60-01271-f005:**
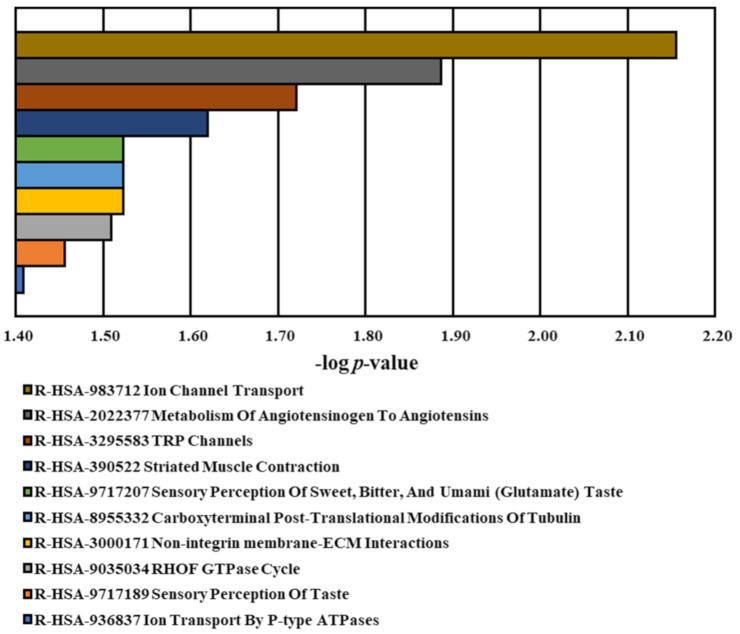
Top 10 significant pathways relative to variants analysis. Genes associated with identified risk variants were characterized using the Reactome database. Selected pathways exhibited with *p*-values < 0.05.

**Figure 6 medicina-60-01271-f006:**
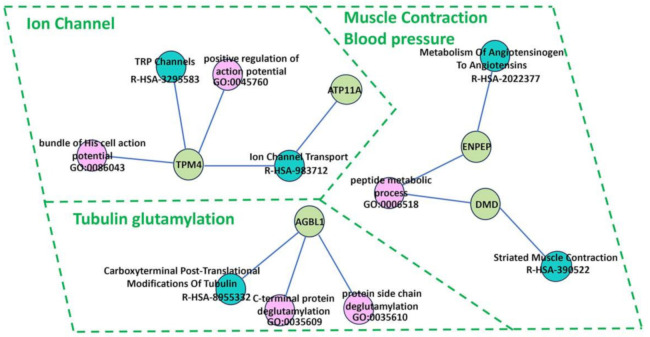
Gene functional and molecular pathway annotation network of AGBL1, ATP11A, DMD, ENPEP, and TPM. The functional annotation network was constructed using GO biological process (green circles) and Reactome pathway (pink circles).

**Table 1 medicina-60-01271-t001:** Demographic characteristics of the participants.

Group	Schizophrenia ^a^ (*n* = 137)		Control (*n* = 25,927)	
Sex				
Male	63	(46.0%)	11,676	(45.0%)
Female	74	(54.0%)	14,251	(55.0%)
Age				
80–89	1	(0.73%)	895	(3.42%)
70–79	5	(3.65%)	2970	(11.36%)
60–69	13	(9.49%)	5682	(21.74%)
50–59	33	(24.09%)	5124	(19.60%)
40–49	40	(29.20%)	4096	(15.67%)
30–39	27	(19.71%)	3425	(13.1%)
20–29	17	(12.41%)	3002	(11.49%)
<20	1	(0.73%)	733	(2.81%)

^a^ Defined according to the ICD-10-CM code F20.0, F20.1, F20.2, F20.5.

**Table 2 medicina-60-01271-t002:** Variants identified from highly significant GWAS results.

CHR	Position	SNP ID	Ref ^a^	Alt ^b^	*p*-Value ^c^	Odds Ratio ± SD	95% CI	Effect Size ^d^	Region	Relative Gene
1	14175667	rs12028521	A	G	6.72 × 10^−6^	0.53 ± 0.14	0.40−0.70	−0.004	intergenic	*PRDM2*; *KAZN*
1	190853292	rs1170702	T	C	6.80 × 10^−6^	2.22 ± 0.18	1.55−3.17	0.000	intergenic	*LINC01720*; *LINC01680*
2	109675309	rs1010840	T	C	7.20 × 10^−6^	1.74 ± 0.12	1.36−2.22	0.014	intergenic	*SOWAHC*; *RGPD6*
4	110552210	rs9997349	A	G	3.62 × 10^−6^	2.18 ± 0.17	1.55−3.05	0.014	intronic	*ENPEP*
8	129221357	rs78144262	G	A	2.14 × 10^−8^	2.24 ± 0.15	1.68−2.99	0.026	ncRNA_intronic	*LINC00977*
13	65637888	rs11619878	G	A	5.71 × 10^−7^	2.25 ± 0.17	1.63−3.13	0.018	intergenic	*LINC00355*; *LINC01052*
13	112867397	rs1556812	C	G	1.87 × 10^−7^	2.50 ± 0.18	1.75−3.57	0.000	intronic	*ATP11A*
15	85947722	rs117010638	A	C	6.07 × 10^−6^	2.12 ± 0.17	1.52−2.96	0.001	intergenic	*MIR548AP*; *AGBL1*
19	49162083	rs11083963	G	A	7.11 × 10^−6^	1.72 ± 0.12	1.36−2.19	0.014	intronic	*TRPM4*
X	31237060	rs12859097	G	T	5.44 × 10^−6^	2.22 ± 0.18	1.56−3.16	0.016	intronic	*DMD*

^a^ Allele in control. ^b^ Allele in case. ^c^ Filter by *p* < 10^−5^. ^d^ Estimated by lasso regression.

**Table 3 medicina-60-01271-t003:** Allele frequencies of variants across multiple large genetic projects.

This Study				TPMI ^a^	Twbank ^b^	1000g ^c^				gnomAD ^d^				
SNP ID	Alt	Case	Control			AFR	AMR	EAS	EUR	AFR	AMR	EAS	FIN	NFE
rs12028521	G	0.234	0.365	0.365	0.371	0.200	0.310	0.350	0.110	0.174	0.258	0.361	0.114	0.109
rs1170702	C	0.128	0.062	0.060	0.059	-	-	-	-	-	-	-	-	-
rs1010840	C	0.383	0.263	0.267	0.264	0.086	0.190	0.240	0.075	0.103	0.168	0.269	0.059	0.080
rs9997349	G	0.146	0.073	0.072	0.073	-	-	-	-	-	-	-	-	-
rs78144262	A	0.215	0.109	0.111	0.119	-	0.099	0.090	0.002	0.002	0.146	0.109	0.013	0.002
rs11619878	A	0.157	0.076	0.078	0.076	0.430	0.410	0.086	0.500	0.443	0.384	0.082	0.501	0.518
rs1556812	G	0.128	0.055	0.058	0.061	0.340	0.300	0.085	0.240	0.285	0.276	0.058	0.266	0.219
rs117010638	C	0.150	0.077	0.076	0.076	-	-	0.090	0.012	0.000	0.000	0.073	0.059	0.004
rs11083963	A	0.577	0.442	0.440	0.430	0.200	0.530	0.390	0.550	0.266	0.596	0.435	0.524	0.560
rs12859097	T	0.180	0.090	0.092	0.094	-	-	-	-	-	-	-	-	-

Abbreviations: SNP, single nucleotide polymorphism; AFR, African; AMR, American; EAS, East Asian; EUR, European; FIN, Finnish; NFE, non-Finnish European. ^a^ Taiwan Precision Medicine Initiative. ^b^ Taiwan Biobank. ^c^ 1000 Genomes global minor allele frequency. ^d^ gnomAD (genomes) allele frequencies.

**Table 4 medicina-60-01271-t004:** GWAS catalog recode of rs11619878.

Variant and Risk Allele	*p*-Value	RAF	Beta	Reported Trait	Trait(s)	Study Accession
rs11619878-G	1 × 10^−7^	0.511	0.163 unit decrease	IgG glycosylation	Serum IgG glycosylation measurement	GCST001848

**Table 5 medicina-60-01271-t005:** GO annotation of variant genes ^a^.

GO_ID	Term	*p*-Value	Gene
GO:0006518	Peptide metabolic process	0.002	*ENPEP*; *DMD*
GO:0035609	C-terminal protein deglutamylation	0.004	*AGBL1*
GO:0035610	Protein side chain deglutamylation	0.004	*AGBL1*
GO:0045760	Positive regulation of action potential	0.004	*TRPM4*
GO:0086043	Bundle of His cell action potential	0.004	*TRPM4*
GO:0086045	Membrane depolarization during AV node cell action potential	0.004	*TRPM4*
GO:1904177	Regulation of adipose tissue development	0.004	*TRPM4*
GO:2000169	Regulation of peptidyl-cysteine S-nitrosylation	0.004	*DMD*
GO:0003081	Regulation of systemic arterial blood pressure by renin-angiotensin	0.005	*ENPEP*
GO:0014819	Regulation of skeletal muscle contraction	0.005	*DMD*
GO:0086016	AV node cell action potential	0.005	*TRPM4*
GO:1904181	Positive regulation of membrane depolarization	0.005	*TRPM4*
GO:0010831	Positive regulation of myotube differentiation	0.006	*ATP11A*
GO:0035608	Protein deglutamylation	0.006	*AGBL1*
GO:0002709	Regulation of T cell mediated immunity	0.008	*TRPM4*
GO:0002002	Regulation of angiotensin levels in blood	0.009	*ENPEP*
GO:0002003	Angiotensin maturation	0.009	*ENPEP*
GO:0018410	C-terminal protein amino acid modification	0.009	*AGBL1*
GO:0098660	Inorganic ion transmembrane transport	0.009	*TRPM4*
GO:0098911	Regulation of ventricular cardiac muscle cell action potential	0.009	*TRPM4*
GO:1902307	Positive regulation of sodium ion transmembrane transport	0.009	*DMD*
GO:0030502	Negative regulation of bone mineralization	0.010	*TRPM4*
GO:0035774	Positive regulation of insulin secretion involved in cellular responseto glucose stimulus	0.010	*TRPM4*
GO:2000651	Positive regulation of sodium ion transmembrane transporter activity	0.010	*DMD*
GO:0035994	Response to muscle stretch	0.011	*DMD*
GO:0001990	Regulation of systemic arterial blood pressure by hormone	0.012	*ENPEP*
GO:0002407	Dendritic cell chemotaxis	0.012	*TRPM4*
GO:0045907	Positive regulation of vasoconstriction	0.012	*TRPM4*
GO:0010460	Positive regulation of heart rate	0.013	*TRPM4*
GO:0010881	Regulation of cardiac muscle contraction by regulation of the release ofsequestered calcium ion	0.013	*DMD*
GO:0036336	Dendritic cell migration	0.013	*TRPM4*
GO:0086012	Membrane depolarization during cardiac muscle cell action potential	0.013	*TRPM4*
GO:0098719	Sodium ion import across plasma membrane	0.013	*TRPM4*
GO:0002724	Regulation of T cell cytokine production	0.014	*TRPM4*
GO:0045823	Positive regulation of heart contraction	0.015	*TRPM4*
GO:0010882	Regulation of cardiac muscle contraction by calcium ion signaling	0.016	*DMD*
GO:0033137	Negative regulation of peptidyl-serine phosphorylation	0.016	*DMD*
GO:0070168	Negative regulation of biomineral tissue development	0.016	*TRPM4*
GO:1901385	Regulation of voltage-gated calcium channel activity	0.016	*DMD*
GO:0010880	Regulation of release of sequestered calcium ion into cytosol by sarcoplasmicreticulum	0.017	*DMD*
GO:0002718	Regulation of cytokine production involved in immune response	0.018	*TRPM4*
GO:0030279	Negative regulation of ossification	0.018	*TRPM4*
GO:0034204	Lipid translocation	0.018	*ATP11A*
GO:0010830	Regulation of myotube differentiation	0.019	*ATP11A*
GO:0032414	Positive regulation of ion transmembrane transporter activity	0.019	*DMD*
GO:0043171	Peptide catabolic process	0.019	*ENPEP*
GO:0060314	Regulation of ryanodine-sensitive calcium-release channel activity	0.019	*DMD*
GO:0098901	Regulation of cardiac muscle cell action potential	0.019	*TRPM4*
GO:0051155	Positive regulation of striated muscle cell differentiation	0.020	*ATP11A*
GO:0086004	Regulation of cardiac muscle cell contraction	0.020	*TRPM4*
GO:0061178	Regulation of insulin secretion involved in cellular response to glucosestimulus	0.021	*TRPM4*
GO:0006942	Regulation of striated muscle contraction	0.022	*DMD*
GO:0019229	Regulation of vasoconstriction	0.022	*TRPM4*
GO:0035633	Maintenance of blood-brain barrier	0.022	*DMD*
GO:0086001	Cardiac muscle cell action potential	0.022	*DMD*
GO:1901019	Regulation of calcium ion transmembrane transporter activity	0.022	*DMD*
GO:0045332	Phospholipid translocation	0.023	*ATP11A*
GO:0045668	Negative regulation of osteoblast differentiation	0.023	*TRPM4*
GO:0055001	Muscle cell development	0.024	*DMD*
GO:0060048	Cardiac muscle contraction	0.025	*DMD*
GO:0042692	Muscle cell differentiation	0.026	*DMD*
GO:0030049	Muscle filament sliding	0.028	*DMD*
GO:0032024	Positive regulation of insulin secretion	0.028	*TRPM4*
GO:0033275	Actin-myosin filament sliding	0.028	*DMD*
GO:0086091	Regulation of heart rate by cardiac conduction	0.029	*TRPM4*
GO:0060047	Heart contraction	0.030	*DMD*
GO:2000649	Regulation of sodium ion transmembrane transporter activity	0.030	*DMD*
GO:0001508	Action potential	0.033	*DMD*
GO:0098655	Cation transmembrane transport	0.033	*TRPM4*
GO:0043604	Amide biosynthetic process	0.038	*DMD*
GO:0045600	Positive regulation of fat cell differentiation	0.038	*TRPM4*
GO:0051279	Regulation of release of sequestered calcium ion into cytosol	0.038	*DMD*
GO:0051289	Protein homotetramerization	0.038	*TRPM4*
GO:0006941	Striated muscle contraction	0.040	*DMD*
GO:0016925	Protein sumoylation	0.040	*TRPM4*
GO:0030500	Regulation of bone mineralization	0.042	*TRPM4*
GO:0007517	Muscle organ development	0.043	*DMD*
GO:0015914	Phospholipid transport	0.043	*ATP11A*
GO:1901565	Organonitrogen compound catabolic process	0.046	*ENPEP*

^a^ Selected based on *p*-value < 0.05.

**Table 6 medicina-60-01271-t006:** Reactome annotation of variant genes ^a^.

Reactome ID	Term	*p*-Value	Gene
R-HSA-983712	Ion channel transport	0.007	*ATP11A*; *TRPM4*
R-HSA-2022377	Metabolism of angiotensinogen to angiotensins	0.013	*ENPEP*
R-HSA-3295583	TRP channels	0.019	*TRPM4*
R-HSA-390522	Striated muscle contraction	0.024	*DMD*
R-HSA-3000171	Non-integrin membrane–ECM interactions	0.030	*DMD*
R-HSA-8955332	Carboxyterminal post-translational modifications of tubulin	0.030	*AGBL1*
R-HSA-9717207	Sensory perception of sweet, bitter, and umami (glutamate) taste	0.030	*TRPM4*
R-HSA-9035034	RHOF GTPase cycle	0.031	*SOWAHC*
R-HSA-9717189	Sensory perception of taste	0.035	*TRPM4*
R-HSA-936837	Ion transport by P-type ATPases	0.039	*ATP11A*

^a^ Selected based on *p*-value < 0.05.

## Data Availability

The data presented in this study are available on request from the corresponding author.

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
