# Peer review of "Identification of Schizophrenia Susceptibility Loci in the Urban Taiwanese Population"

_medicina, 2024, doi:10.3390/medicina60081271_

Round 1

Reviewer 1 Report

Comments and Suggestions for Authors

1-)the following sentence maybe not necessary in the abstract

 Genomic DNA was ex- tracted and genotyped using the Taiwan Precision Medicine array.

2-)chi square test information is also not necessary in the abstract.

3-)you can also add more updated references.

4-)you can also add following reference after the following sentence as article mentions cognitive problems are pervasive features in patients with schizophrenia.

Cognitive deficits in working memory, executive function, and processing speed, alongside comorbid substance use disorder and metabolic disease are also recognized aspects of the condition (Wang et al., 2022)

Wang, D. M., Zhu, R. R., Tian, Y., Uludag, K., Chen, J. J., Zhou, H. X., ... & Zhang, X. Y. (2022). Association between MnSOD Activity and Cognitive Impairment in Unmedicated First-Episode Schizophrenia: Regulated by MnSOD Ala-9Val Gene Polymorphism. Antioxidants, 11(10), 1981. 5-)if possible,you can mention effect size in the study. 6-)you may consider removing people aged more than 90. however, if it is too much work to do. its fine. you can mention it as one of the limitations of the study. 7-)please mention why 5 participants dont have age record. 8-)you can consider sharing other 280 177 variants in the supplementary files. I am not sure it was mentioned. Following chi-squared testing,  280,177 variants were detected among TSGH TPMI participants. Among these, ten highly significant variants were 272
identified based on a p-value < 10-5 (upper red line): rs78144262, rs9997349, rs1010840, rs11083963, rs11619878, rs1170702, 273, rs117010638, rs12028521, rs12859097, and rs1556812. 9-)please mention that it is from the previous study. Notably, among these, the genetic variant rs11619878 has exhibited an association with serum IgG glycosylation levels, indicative of a potential role of glycosylation in schiz-ophrenia 10-) add more limitations related to your study.

Comments on the Quality of English Language

Language can be improved. Chatbot use can be disclosed.

Author Response

Dear Editor and Reviewers:  

We thank for the editors and the reviewer for thoroughly reading our manuscript and providing detailed valuable comments and suggestions. Your comments are really insightful and helpful, and we do honestly agree with them. All your concerns have been taken seriously and incorporated in the revised paper, making the manuscript clearer, more compelling and broader. We have made point-by-point responses and revisions in the following tables.

Reviewer 1#

Comments

Responses

The following sentence maybe not necessary in the abstract.

Genomic DNA was extracted and genotyped using the Taiwan Precision Medicine array.

We thank the reviewer for the insightful comment. We have revised the Abstract accordingly.

chi square test information is also not necessary in the abstract.

We appreciate the reviewer’s suggestion. The Abstract has been updated to exclude this information.

you can also add more updated references.

We thank the reviewer for the insightful comments. We agree with your suggestion and have added several updated references to the manuscript.

References:

3. Wang, D.M., et al., Association between MnSOD Activity and Cognitive Impairment in Unmedicated First-Episode Schizophrenia: Regulated by MnSOD Ala-9Val Gene Polymorphism. Antioxidants (Basel), 2022. 11(10).

9. Ma, J., et al., Association Between the COMT Val158Met Polymorphism and Antipsychotic Efficacy in Schizophrenia: An Updated Meta-Analysis. Curr Neuropharmacol, 2021. 19(10): p. 1780-1790.

34. Zong, P., et al., TRP Channels in Excitotoxicity. Neuroscientist, 2024: p. 10738584241246530.

you can also add following reference after the following sentence as article mentions cognitive problems are pervasive features in patients with schizophrenia.

Cognitive deficits in working memory, executive function, and processing speed, alongside comorbid substance use disorder and metabolic disease are also recognized aspects of the condition (Wang et al., 2022)

Wang, D. M., Zhu, R. R., Tian, Y., Uludag, K., Chen, J. J., Zhou, H. X., ... & Zhang, X. Y. (2022). Association between MnSOD Activity and Cognitive Impairment in Unmedicated First-Episode Schizophrenia: Regulated by MnSOD Ala-9Val Gene Polymorphism. Antioxidants11(10), 1981.

We thank the reviewer for the insightful comments. We agree with your suggestion and have added the reference for cognitive concerns in patients with schizophrenia in the introduction.

Line 45: “Cognitive deficits in working memory, executive function, and processing speed, alongside comorbid substance use disorder and metabolic disease are also recognized aspects of the condition [1, 3].

References:

3. Wang, D.M., et al., Association between MnSOD Activity and Cognitive Impairment in Unmedicated First-Episode Schizophrenia: Regulated by MnSOD Ala-9Val Gene Polymorphism. Antioxidants (Basel), 2022. 11(10).

if possible,you can mention effect size in the study.

We appreciate your suggestions. The effect size was estimated using lasso regression. The method of effect size calculation and the results are described and presented in lines 239-248, lines 314-318, and Table 2.

Line 239: “Variants exhibiting high significance (p-values < 1 × 10-5) from the chi-squared test were selected for further characterization by annotation, such as genetic structure location, relative gene symbols, population frequencies, and biological functions. To estimate variant effect sizes, lasso regression (Least Absolute Shrinkage and Selection Operator) [16] supported by PLINK 1.9 was applied:

1.     Preparation of Data: The phenotype and genotype data prepared were the same as those used in the chi-squared test.

2.     Loading Data into Lasso Model: Phenotype and genotype data were loaded into the lasso model, with the lambda parameter set to 0.95.

3.     Calculation of Effect Size: The effect size was calculated using lasso regression.

Line 314: “The effect size of these variants indicates that rs78144262 had the strongest association influencing schizophrenia. Other variants, including rs11619878, rs12859097, rs11083963, rs1010840, and rs9997349, showed similar effects. In contrast, variants rs12028521 and rs117010638 had weaker effects, while rs1170702 and rs1556812 showed no effect.

you may consider removing people aged more than 90. however, if it is too much work to do. its fine. you can mention it as one of the limitations of the study.

We greatly appreciate your comment. All participants aged over 90 have been removed, and the association re-analysis has been completed. The revised data are presented in new Figures 1, 2, and 3, as well as Tables 1, 2, and 3. The relevant descriptions have also been revised with these results in lines 301-305.

Line 301: “This study included 137 participants with schizophrenia and 25,927 control partic-ipants (Table 1). After conducting the schizophrenia-genetic association analysis, 280,153 SNPs were selected, all of which passed the filters for SNP call rate, minor allele frequency, and Hardy-Weinberg equilibrium.

please mention why 5 participants dont have age record.

Considering these participants lost age information and the sample size is too small to strongly affect the association analysis, we decided to remove these participants along with those aged over 90. The presentation of re-analysis results is described above in point 6.

you can consider sharing other 280 177 variants in the supplementary files. I am not sure it was mentioned. Following chi-squared testing,  280,177 variants were detected among TSGH TPMI participants. Among these, ten highly significant variants were 272 identified based on a p-value < 10-5 (upper red line): rs78144262, rs9997349, rs1010840, rs11083963, rs11619878, rs1170702, 273, rs117010638, rs12028521, rs12859097, and rs1556812.

Following your kind suggestion, we listed all variants obtained from chi-squared testing in Supplementary Table 1. Additionally, 280,153 variants, including these ten highly significant ones obtained from re-analysis according to points 6 and 7, were included.

please mention that it is from the previous study. Notably, among these, the genetic variant rs11619878 has exhibited an association with serum IgG glycosylation levels, indicative of a potential role of glycosylation in schizophrenia.

We thank the reviewer for the insightful comments. We revised the paper as per your suggestion and added another paragraph in the Discussion.

Line 516: “Notably, among these, the genetic variant rs11619878 has exhibited an association with serum IgG glycosylation levels, as indicated by the investigation in the GWAS Catalog database [24], suggesting a potential role of glycosylation in schizophrenia.

add more limitations related to your study.

We thank the reviewer for the insightful comments. We agree with your opinion and have revised the Limitations section.

Line 693: “First, our study failed to demonstrate some variants that were found in previous large-scale genetic studies in schizophrenia. The results reported herein should be in-terpreted with caution because the generalizability of the findings is restricted by the small sample size and unmeasured confounding factors in the population demographics. Second, the present study investigated common genetic variations with minor allele frequency < 0.05 excluded. Therefore, there may be rare variants of importance related to schizophrenia that fall outside the scope of this study.

Line 705: “Additionally, while our study identified many variants and relative genes potentially comorbid with schizophrenia, the cross-sectional design limits the ability to determine whether the shared genetic influences found in this and previous studies are causal factors for the comorbidity of schizophrenia. Social and environmental factors are likely to play at least as large a role as genetic ones in the development and co-occurrence of schizophrenia.

Reviewer 2 Report

Comments and Suggestions for Authors

In reviewing  manuscript, I recommend evaluating the adequacy of the sample size and the appropriateness of participant selection in relation to the study's objectives. Further scrutiny of the statistical analyses and tests used is warranted to ensure they adequately address the research questions posed. Additional details should be provided on the genotyping platforms used and the methods employed to explore genotype-phenotype correlations. The literature review section could benefit from an update to include the most recent and comprehensive studies, thereby ensuring a robust framework for discussion. Comparisons of the study results with existing literature should be made explicit, particularly where discrepancies arise, and suggestions for integration with existing theories should be provided. To enhance the generalizability of the findings, conducting similar studies across diverse populations is advised. Future research should also consider exploring new variables, such as environmental factors or epigenetic changes. It is crucial to clearly articulate the methodological limitations of the study and discuss their potential impacts on the results. Addressing limitations related to the applicability and interpretation of the findings is essential, along with providing strategies to overcome these issues. Lastly, I encourage the inclusion of the latest research findings in the literature review to maintain the manuscript’s relevance and ensure it reflects current trends and knowledge in the field.

Comments on the Quality of English Language

Minor editing of English language required

Author Response

Dear Editor and Reviewers:  

We thank for the editors and the reviewer for thoroughly reading our manuscript and providing detailed valuable comments and suggestions. Your comments are really insightful and helpful, and we do honestly agree with them. All your concerns have been taken seriously and incorporated in the revised paper, making the manuscript clearer, more compelling and broader. We have made point-by-point responses and revisions in the following tables.

Reviewer 2#

Comments

Responses

In reviewing  manuscript, I recommend evaluating the adequacy of the sample size and the appropriateness of participant selection in relation to the study's objectives. Further scrutiny of the statistical analyses and tests used is warranted to ensure they adequately address the research questions posed. Additional details should be provided on the genotyping platforms used and the methods employed to explore genotype-phenotype correlations.

  The literature review section could benefit from an update to include the most recent and comprehensive studies, thereby ensuring a robust framework for discussion. Comparisons of the study results with existing literature should be made explicit, particularly where discrepancies arise, and suggestions for integration with existing theories should be provided.

To enhance the generalizability of the findings, conducting similar studies across diverse populations is advised. Future research should also consider exploring new variables, such as environmental factors or epigenetic changes.

It is crucial to clearly articulate the methodological limitations of the study and discuss their potential impacts on the results. Addressing limitations related to the applicability and interpretation of the findings is essential, along with providing strategies to overcome these issues. Lastly, I encourage the inclusion of the latest research findings in the literature review to maintain the manuscript’s relevance and ensure it reflects current trends and knowledge in the field.

We appreciate your kind suggestions. For evaluating sample size, the allele frequency of the control group in this study is close to that of the TPMI and Taiwan Biobank, which includes about 500,000 Taiwanese participants from 16 local medical centers. This sample size is sufficiently large for the allele frequency to follow a normal distribution and be representative of healthy participants enrolled in the Taiwan Biobank (see Table 3). Additionally, we excluded control participants enrolled from neurological and psychiatry departments to increase the study’s precision. For statistical analysis, we revised our association analysis to include standard deviation, 95% CI range, and variant effect size estimation, as presented in revised Table 2. The methods of the genotyping platform and genotype-phenotype correlations are revised in lines 167-178 and lines 211-236.

  We have updated the literature review to include the most recent and comprehensive studies. This ensures a robust framework for the discussion section in the revised manuscript.

  To further validate these results, we acknowledge the importance of considering different populations and environmental factors. We plan to incorporate these comments in our future studies.

  We have revised the limitations section and added several updated references to maintain the manuscript’s relevance and ensure it reflects current trends and knowledge in the field.

Round 2

Reviewer 2 Report

Comments and Suggestions for Authors

The requested revisions have been made. However, Figures 1, 2, 3, 4, 5, and 6 need to be rearranged. In their current form, they are too large and incomprehensible. Once these adjustments are made, the article will be acceptable for publication.

Comments on the Quality of English Language

Minor editing of English language required